# Hierarchical semantic composition of biosimulation models using bond graphs

**Niloofar Shahidi**[ID][1]*, **Michael Pan**[ID][2,3], **Soroush Safaei**[ID][1], **Kenneth Tran**[ID][1], **Edmund J. Crampin**[ID][2,3], **David P. Nickerson**[ID][1]

**1** Auckland Bioengineering Institute, The University of Auckland, Auckland, New Zealand, **2** Systems Biology Laboratory, School of Mathematics and Statistics, and Department of Biomedical Engineering, University of Melbourne, Melbourne, Victoria, Australia, **3** ARC Centre of Excellence in Convergent Bio-Nano Science and Technology, Faculty of Engineering and Information Technology, University of Melbourne, Melbourne, Victoria, Australia

* nsha457@aucklanduni.ac.nz

**Data Availability Statement:** The reference model data are available from: https://models. physiomeproject.org/workspace/4ac/file/ feb8451c388983536ab9e20d1752ff3659977da8/

## Abstract

Simulating complex biological and physiological systems and predicting their behaviours under different conditions remains challenging. Breaking systems into smaller and more manageable modules can address this challenge, assisting both model development and simulation. Nevertheless, existing computational models in biology and physiology are often not modular and therefore difficult to assemble into larger models. Even when this is possible, the resulting model may not be useful due to inconsistencies either with the laws of physics or the physiological behaviour of the system. Here, we propose a general methodology for composing models, combining the energy-based bond graph approach with semantics-based annotations. This approach improves model composition and ensures that a composite model is physically plausible. As an example, we demonstrate this approach to automated model composition using a model of human arterial circulation. The major benefit is that modellers can spend more time on understanding the behaviour of complex biological and physiological systems and less time wrangling with model composition.

## Author summary

Biological and physiological systems usually involve multiple underlying processes, mechanisms, structures, and phenomena, referred to here as sub-systems. Modelling the whole system every time from scratch requires a huge amount of effort. An alternative is to model each sub-system in a modular fashion, *i.e.*, containing meaningful interfaces for connecting to other modules. Such modules are readily combined to produce a whole-system model. For the combined model to be consistent, modules must be described using the same modelling scheme. One way to achieve this is to use energy-based models that are consistent with the conservation laws of physics. Here, we present an approach that achieves this using bond graphs, which allows modules to be combined faster and more efficiently. First, physically plausible modules are generated using a small number of template modules. Then a meaningful interface is added to each module to automate

main_ADAN-86.cellml/ All the model files are available on GitHub: https://github.com/Niloofar-Sh/ADAN-86-Bond-Graph-Model-Composition.

**Funding:** NS is supported by an Aotearoa Fellowship to DPN from the Aotearoa Foundation. MP is supported by the Australian Research Council Centre of Excellence in Convergent Bio-Nano Science and Technology (project number CE140100036) (http://purl.org/au-research/grants/arc/CE140100036) to EJC. SS is supported by an Aotearoa Fellowship from the Aotearoa Foundation. KT is supported by a Marsden Fast-Start grant (UOA1703) from the Royal Society of New Zealand (https://www.royalsociety.org.nz) and a Sir Charles Hercus Health Research Fellowship (21/116) from the Health Research Council of New Zealand (https://gateway.hrc.govt.nz/funding/career-development-awards/2021-sir-charles-hercus-health-research-fellowship). EJC is supported by the Australian Research Council Centre of Excellence in Convergent Bio-Nano Science and Technology (project number CE140100036) (http://purl.org/au-research/grants/arc/CE140100036). DPN is supported by an Aotearoa Fellowship from the Aotearoa Foundation and the Center for Reproducible Biomedical Modeling P41 EB023912/EB/NIBIB NIH HHS/United States (https://projectreporter.nih.gov/project_description.cfm?projectnumber=5P41EB023912-03). The funders had no role in study design, data collection and analysis, decision to publish, or preparation of the manuscript.

**Competing interests:** The authors have declared that no competing interests exist.

connection. This approach is illustrated by applying this method to an existing model of the circulatory system and verifying the results against the reference model.

## Introduction

Mathematical models have long been used to study biological systems and predict their behaviours [1, 2]. However, the expansion of biological models to bridge between cells and organs demands reusable and comprehensible existing models to avoid the repetitive work of recreating them [3, 4]. Constructing a complex model by composing various well-defined sub-models (modularisation) helps researchers handle this complexity [5], facilitates model composition and promotes model sharing [6–8].

Mathematical modelling in the context of biology was first intended to simplify the analysis of biological and physiological processes and systems. Such models are generally applicable in the context in which they were developed, which determines how complicated a model should be [9, 10]. While such single-purpose models are often computationally efficient, they cannot be utilised to simulate different biological conditions [10]. This occurs because single-purpose models only give the investigator an insight into the specific data from which they were derived, and are unable to predict beyond that given knowledge [11, 12]. Complex, fully detailed models may be valid under more physiological conditions, but are often challenging to comprehend and simulate [4]. In the end, we need a compromise between complexity and simplicity while capturing the main physical features of a system.

In recent years, as the Physiome (www.physiomeproject.org) and Virtual Physiological Human (VPH) (www.vph-institute.org) projects have demonstrated, initial steps have been taken to construct more realistic models able to describe almost every system in the body (from cell to organs) [13, 14]. By assembling these models (model composition), one can construct a model of the whole or part of the human body. However, many computational models are not expressed in a uniform format, making it difficult—if not impossible—to assemble them [15]. Most existing mathematical modelling methods are not flexible enough for modularisation and require significant editing of the equations and code after recombination the modules [16]. Establishing a link between the physiological processes and biological systems at any level of operation in the body requires multiscale modelling [17, 18]. This calls for a standardised and general-purpose platform, which once implemented, can help scientists conduct *in silico* experiments and examine hypotheses on a virtual body model [19].

Throughout the natural world, *energy* is always conserved [20]. Therefore, if energy is selected as the exchange variable in the models, especially in composing multiple models across different domains of physics, it ensures that all the individual models are thermodynamically and physically consistent [21, 22]. Bond graphs provide an energy-based and general-purpose modelling framework that ensures models obey thermodynamic and physical principles [22, 23]. Bond graphs were invented by Henry Paynter and were initially meant to be used in mechanical systems [24]. In bond graphs, all systems are reduced to a graphical representation in which physical components are connected by a network of bonds (graphically represented as half-arrows) and junctions. Bonds carry two physical co-variables: potential (in Joule per quantity) and flow (in quantity per second). Noting that the product of potential and flow is power (in Joules per second), bonds are energy-conserving. Junctions either share a common potential (shown by '0') or share a common flow (shown by '1'). These notations are analogous to the Kirchhoff's laws in electrical circuits, equilibrium of forces in mechanical systems, and stoichiometry balance in chemical reactions [25]. Since these

conservation laws in different domains follow the same mathematical principles, they can be represented by generalised equations using bond graphs.

The application of bond graphs to biochemical processes was proposed by Oster et al. [26] and has recently been advanced by Gawthrop and Crampin [4, 22]. The bond graph description of a biophysical model produces components that have physical and biophysical interpretations, assisting a better understanding of the system. Once each module is represented in bond graph form, correct coupling between modules is straightforward, and the resultant model is itself a bond graph and hence physically consistent [6].

Various software tools have been developed to construct bond graph models. ENPORT was one of the first bond graph simulation languages written in the early seventies by Rosenberg. ENPORT was produced to support mechatronic modelling and was improved later in ENPORT MB [27]. Graphics-based packages such as 20-sim [28], Dymola [29] (based on the open Modelica modeling language), and Wolfram System Modeler [30] (based on the Modelica language) enable users to construct models by selecting bond graph components from a library and draft a graphical display of models. SYMBOLS 2000 [31] allows users to incorporate sophisticated sub-models. Granda et al. [32] proposed a bond graph modelling software—Computer Aided Modelling Program with Graphical Input (CAMP-G) whose generated code from a schematic model is exported to different simulation environments such as MATLAB and ACSL. Within MATLAB Simulink, users are able to assemble components from a variety of physical domains using Simscape [33] (a physical modelling toolbox). Complex components and analysis techniques can be utilised from its add-on products. Almost all the available bond graph software are restricted to predefined components and isolated environments. Bond-GraphTools, recently developed by Cudmore et al. [34], is an open source python library for building symbolic bond graph models. It can be utilised together with other python libraries and includes biochemical components to readily construct biochemical networks. Bond-GraphTools is accessible on GitHub: https://github.com/BondGraphTools.

Endeavours to automate model composition have led to a number of proposed composition platforms, standards, and outlines [15, 35–38], each suitable for a specific set of modelling tasks. Automating this procedure aids the investigators with faster and more reliable model composition in which the linking points between the modules can be automatically detected. However, at present, many biological models do not consider fundamental physical and thermodynamic principles [23], often violating conservation of mass and energy in the underlying equations [39]. Together these produce inconsistencies when models are composed [19]. Semantically enriched bond graph modules would enable the automation of model composition, as each module can be confidently linked to others via common annotated components [38]. There has been a consensus among the Physiome Project and the VPH to label the mathematical contents of the computational models with meaningful semantics, known as annotations [40]. In this way, not only do models contain biological knowledge that is readily interpretable by everyone, but also they become accessible and reusable [14, 40].

Representing the modules in the Cellular modelling Markup Language (CellML) format, meets these standards [5]. CellML is a machine-readable XML-based language which relies on modular modelling and allows reusing components from other models which facilitates model construction. It also reads and runs the simulation for the models containing annotations.

CellML models can be merged using SemGen (available from: https://github.com/SemBioProcess/SemGen/releases). SemGen is a free Java-based application designed for annotating, merging, and extracting biosimulation models encoded in CellML [41], Systems Biology Markup Language (SBML) [42] and JSim's Mathematical Modelling Language (MML) [43, 44]. It generates the semantic annotations as Resource Description Framework (RDF) triples (https://www.w3.org/RDF) and adds them to the model code. Unlike other composition

approaches that rely on pre-defined module interfaces for coupling, SemGen uses a 'white box' approach in which any element of the modules can be selected as joint components by constructing required mappings between the module elements [45]. In contrast, for example, in 'black box' compositions, the internal components of the blocks are hidden. Only the input/output variables are available for the user for coupling the modules [3]. One of the disadvantages of this approach in the biological context is that usually the majority of the entities in a model are potentially capable of being considered as coupling ports. Therefore, using the 'black box' configuration is not compatible with our modelling purposes.

Here, we demonstrate a general method for semantics-based automated model composition using bond graphs, which enables rapid construction of whole body multiscale models. To demonstrate this, we construct and combine simplified, reusable bond graph modules for the Anatomically Detailed Arterial Network (ADAN) open-loop circulatory model based on existing work by Safaei et al. [25, 46]. The ADAN model, first mathematically developed as a partial differential equation (PDE) model by Watanabe et al. [47], anatomically and physiologically describes the arterial network in terms of segments and branches in which blood flow is simulated. Safaei et al. have since developed physically-plausible lumped parameter models based on this work. The ADAN model for the arterial network for the entire body is called ADAN closed-loop whereas the reduced version of this model is called ADAN open-loop model, which represents the arterial network of the body except for the cerebral system. To generate the ADAN open-loop model, we employ SemGen's semantics-based merging tool to capture the common entities across the modules and provide a list of likely variables for composition. Simulation results produced by the assembled model and comparisons to the ADAN open-loop bond graph model are presented and discussed, along with possible improvements to the semantics-based model composition approach.

## Materials and methods

This section discusses the composition of CellML models using SemGen's merging tool to reconstruct the ADAN open-loop model of the arterial system. Particularly, we will show that we only need three templates to generate the 86 vessel segments of the ADAN open-loop model. Later, we will demonstrate the potential for reuse in the case of having similar sections in a system (as in the right and left limbs). Ultimately, the quantities for all the biological entities can also be mapped to the model variables using the SemGen merger tool. Our approach is summarised in Fig 1, illustrating the flowchart of the modelling procedure. SemGen annotator and merger tools are discussed further in [45].

### Semantic model composition using SemGen

SemGen has been used to integrate several mathematical models by treating them as modules [43, 48]. However, SemGen does not currently have the capability to edit the equations in the models being merged [49]. Thus, conflicts may arise while coupling the mathematical models, requiring post-merging adjustments in equations. Often, these post-merging modifications are error-prone, prolonged [48, 50] and require a context-specific knowledge of the underlying equations. Thus, although SemGen has facilitated mathematical model composition, it is still reliant on the modeller's decisions to produce an integrated biological model which produces feasible results [43, 48, 50, 51].

To avoid this, we propose using the bond graph approach for creating the modules. Describing biological and physiological systems in terms of bond graphs in CellML allows us to take the advantage of both the energy-based and hierarchical nature of the bond graph approach and the convenient semi-automated merging tool of SemGen. Existing merging

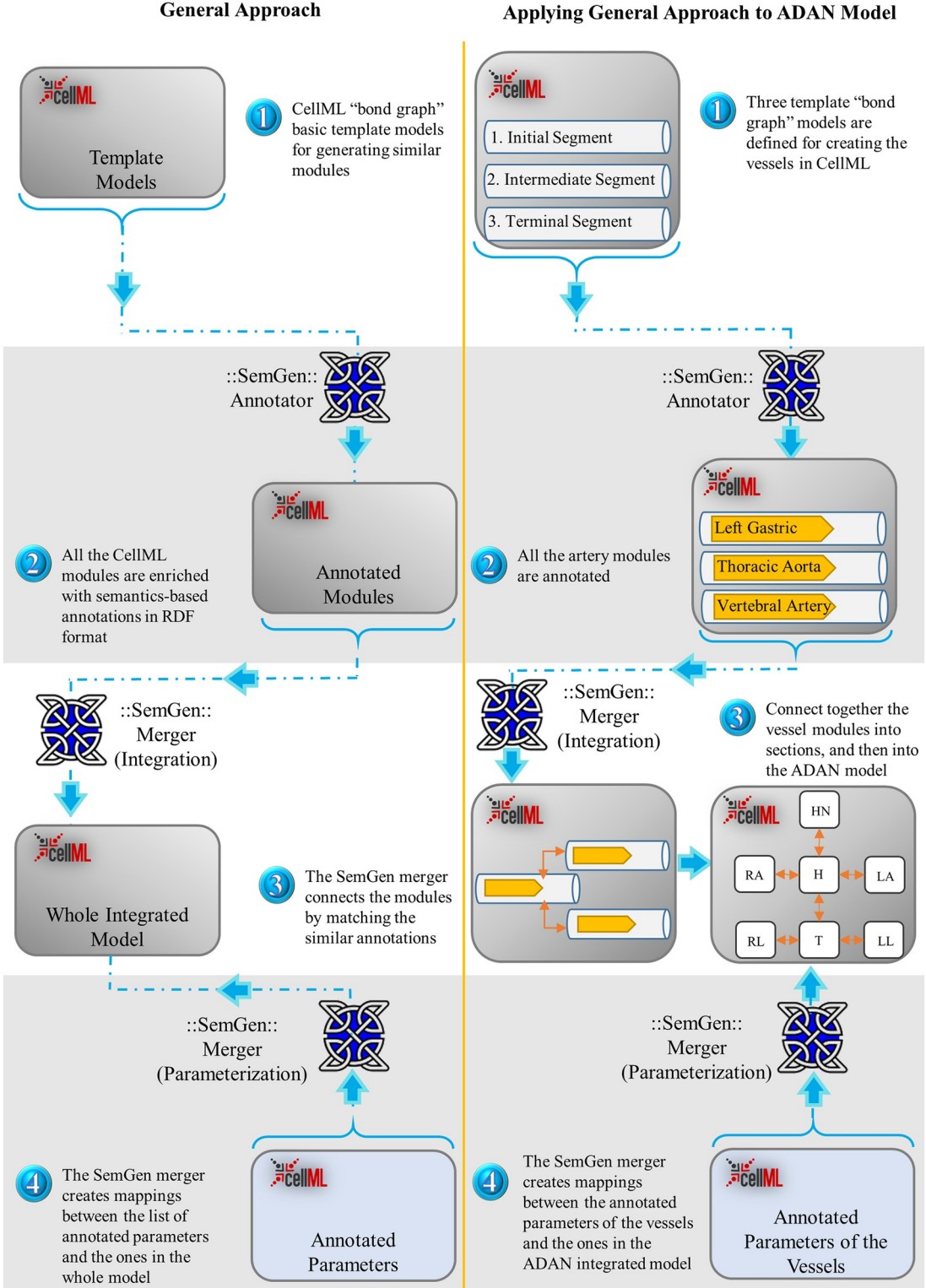

**Fig 1. Flowchart of the semantics-based automated model composition method.** The left column illustrates the general approach in the integration of bond graph models in CellML using SemGen, and the right column shows the same procedure specific to the ADAN circulation model. In both approaches, the bond graph models are created based on template models (1), and annotated using the SemGen Annotator (2). Later, depending on whether the system has similar larger compartments or any symmetry, the SemGen merger tool joins the modules in a number of steps (3). The amounts for the symbolic model parameters are allocated from a CellML file containing the annotated parameters and their values (4). In stage (3): HN: Head and Neck, H: Heart, RA: Right Arm, LA: Left Arm, T: Trunk, RL: Right Leg, LL: Left Leg.

tools like SemGen do not consider physical constraints, while the bond graph framework imposes these constraints to the system. Consequently, by defining the CellML modules using bond graphs, the network structure allows conservation equations to be systematically updated when merging models. This allows us to avoid manually mapping the relationships between the sub-models in CellML. Furthermore, representing the bond graph models in CellML also lets us create mappings via SemGen between different variables and parameters, thus avoiding multiple definitions of the same physiological parameter or variable.

**Human arterial network model.** Safaei et al. [25] developed a detailed CellML model of the human arterial network using bond graphs, based on the ADAN model. The model was created by defining manual mappings between different types of bond graph vessel models. For a large model, these manual mappings are time-consuming and require a significant amount of work. Here, we demonstrate that the same can be achieved in an automated manner using the SemGen merger tool and intrinsic hierarchical properties of bond graphs. We also demonstrate that adding an auxiliary variable to each bond graph module in CellML enables coupling of modules via SemGen. Thereby, the creation of complex models becomes simpler, systematic, hence time-efficient, and less error-prone.

Each vessel segment can be modelled in bond graphs using $R$, $C$, and $I$ components, representing the viscous resistance, vessel wall compliance, and mass inertial effect in fluid mechanics, respectively [25]. Here,

- The potential variable is $u$, the energy density or pressure (J/m$^3$). The flow variable is $v$, the fluid volumetric flow (m$^3$/s);

- Segment compliances are defined using $C$ components, given by the constitutive relation $u = q/C$, where $q$ (m$^3$) is the excess volume caused by dilation of the segment and $C$ is the vessel wall compliance;

- Inertial storage is defined using an $I$ component with a constitutive relation $u = Ia$, where $a = \frac{dv}{dt}$ is the flow rate and $I$ is the mass inertial effect;

- $R$ represents the viscous resistance with a constitutive relation $u = Rv$.

The relationships between the bond graph elements are determined by the energy conservation laws in the network structure and the constitutive relations within components, which make bond graph models physically coherent. The two conjugate variables indicate the flow of energy between the bond graph elements; 'potential' and 'flow' which enable the bidirectional flow of information [52, 53]. A more extensive presentation of bond graph theory, and several examples can also be found in [54, 55].

The parameters of each segment were calculated in terms of vessel properties using the equations:

$$R = \frac{8vl}{\pi r^4} \tag{1}$$

$$I = \frac{\rho l}{\pi r^2} \tag{2}$$

$$C = \frac{2\pi r^3 l}{Eh}. \tag{3}$$

The biological and geometric interpretations of the arterial parameters used in Eqs (1), (2) and (3) are given in Table 1.

**Table 1. Vessel properties description.**

| Vascular Properties | | | |
|---|---|---|---|
| Parameter | Definition | Value | Unit |
| $v$ | Blood viscosity | 0.004 | J.s.m$^{-3}$ |
| $\rho$ | Blood density | 1050 | J.s$^2$.m$^{-5}$ |
| $E$ | Young's modulus | $0.4 \times 10^6$ | J.m$^{-3}$ |
| $l$ | Length | (Segment specific) | m |
| $r$ | Radius | | m |
| $h$ | Wall thickness | $r(ae^{br} + ce^{dr})$ | m |
| $a$ | Fitting coefficient | 0.2802 | – |
| $b$ | Fitting coefficient | −505.3 | m$^{-1}$ |
| $c$ | Fitting coefficient | 0.1324 | – |
| $d$ | Fitting coefficient | −11.14 | m$^{-1}$ |

The wall thickness constants ($a$, $b$, $c$, $d$) are mathematical fitting parameters and are considered to be the same for all the vessels.

The dependency of the vessel wall thickness and its radius is logarithmic, as discussed in [56] and later in [57] and [25].

## Template modules

Vessels are represented by generic template bond graph modules based on their mechanical properties. Each non-branching vessel segment is composed of a parallel $C$, an $R$ and $I$ in series, connected by bonds (Fig 2). To emphasise the concept of sharing common potential and common flow in '0' and '1' junctions, we denote them as '0 : $u$' and '1 : $v$', respectively, in which $u$ stands for potential and $v$ stands for flow.

Depending on whether a vessel is located adjacent to the source of flow ($S_f$), at a terminal point, or in the middle of the vessels network (bifurcating or not), we modify the original configuration of Fig 2 and define three types of modules: initial, intermediate, and terminal. These three types of bond graph modules are described in Fig 3. $R_A$ and $R_B$ in Fig 3A and 3B account for the viscoelastic effect of the vessel wall.

To ensure the conservation of mass and energy, potential and flow must obey conservation laws, *i.e.*, the sum of all the energy flows at each junction is zero [55] (*i.e.*, energy is neither created nor destroyed at a junction):

$$\sum_{i=1}^{n} u_i.v_i = 0 \tag{4}$$

where $i$ indicates the number of bonds impinging on a junction. Depending on the type of junction, one co-variable is considered fixed in the conservation equations. In a '0 : $u$' junction the potential is fixed; and in a '1 : $v$' junction the flow is fixed. Eqs (5) and (6) are the conservation relations for '0 : $u$' and '1 : $v$' junctions, respectively.

$$\begin{cases} u_1 = u_2 = \ldots = u_n, \\ \sum_{i=1}^{n} v_i = 0 \end{cases} \tag{5}$$

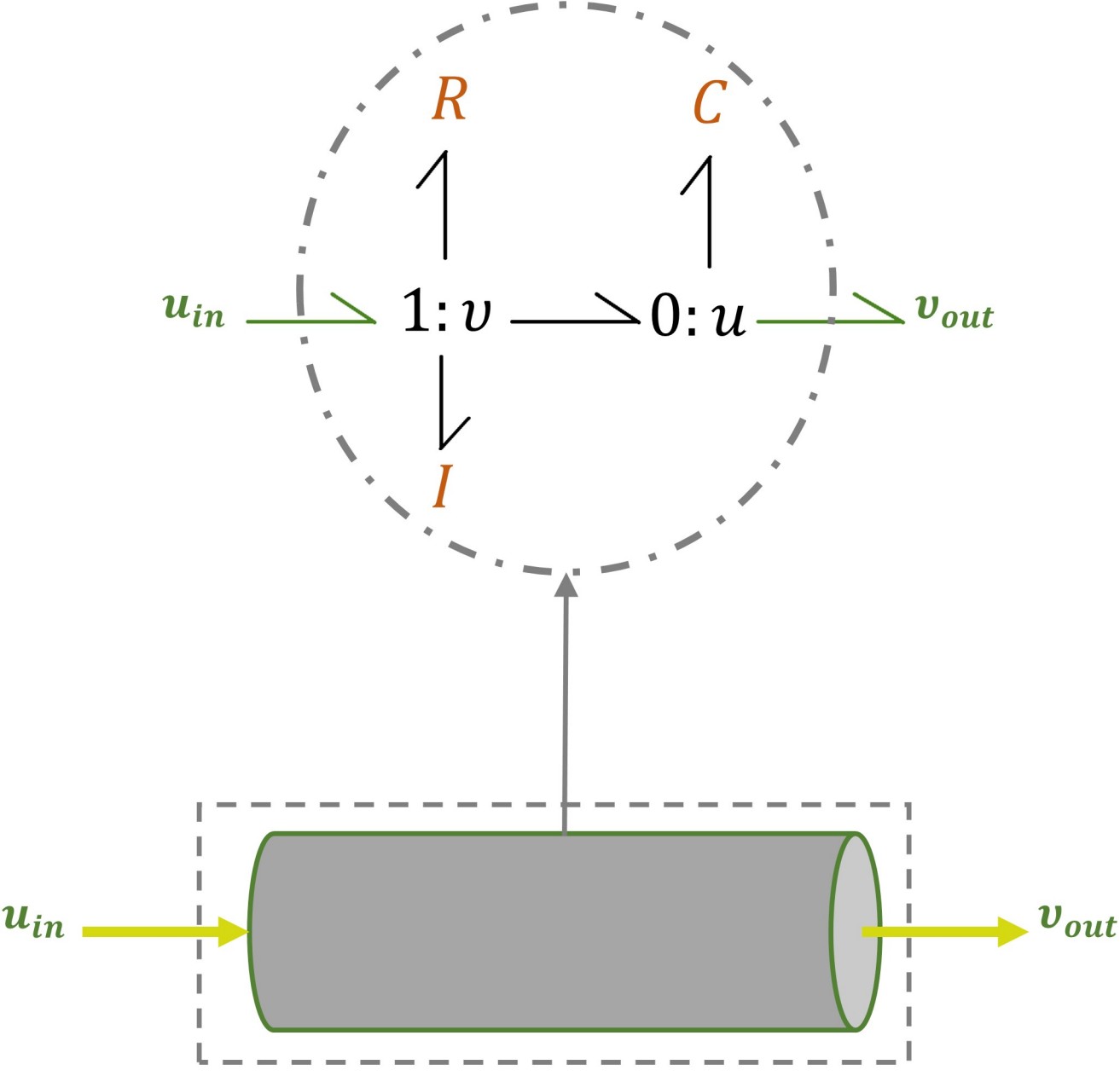

**Fig 2. Bond graph representation of a non-branching vessel.** $C$, $R$ and $I$ components show the mechanical properties of a vessel segment.

$$\begin{cases} v_1 = v_2 = \ldots = v_n, \\ \sum_{i=1}^{n} u_i = 0 \end{cases} \quad (6)$$

The terminal junctions in each module are selected for coupling with other modules. When two modules are coupled, and a bond is added to a junction, there is consequently a change in the conservation equations of that junction. For instance, consider the modules in Fig 3A and 3B. The conservation equations at their terminal junctions would be as in Fig 4.

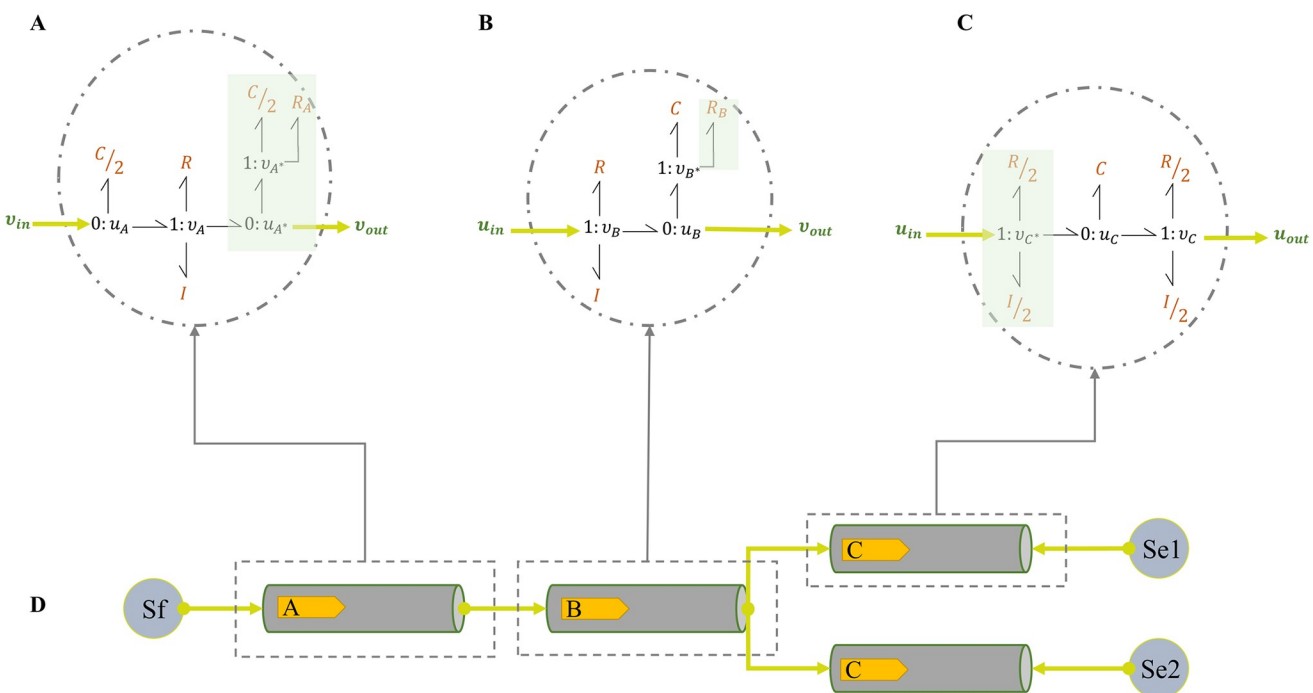

**Fig 3. Required bond graph template modules for the vessels in the ADAN open-loop model.** Depending on the type and location of a vessel, three template modules can be proposed: (A) initial segment; (B) intermediate segment; (C) terminal segment; (D) the schematic of the segments connections. $S_f$ and $\{Se_1, Se_2\}$ are sources of flow and pressure in bond graphs which are *cardiac output* and *venous pressure* in the ADAN open-loop model [25]. The components in green boxes are the added sections to the original bond graph configuration of Fig 2.

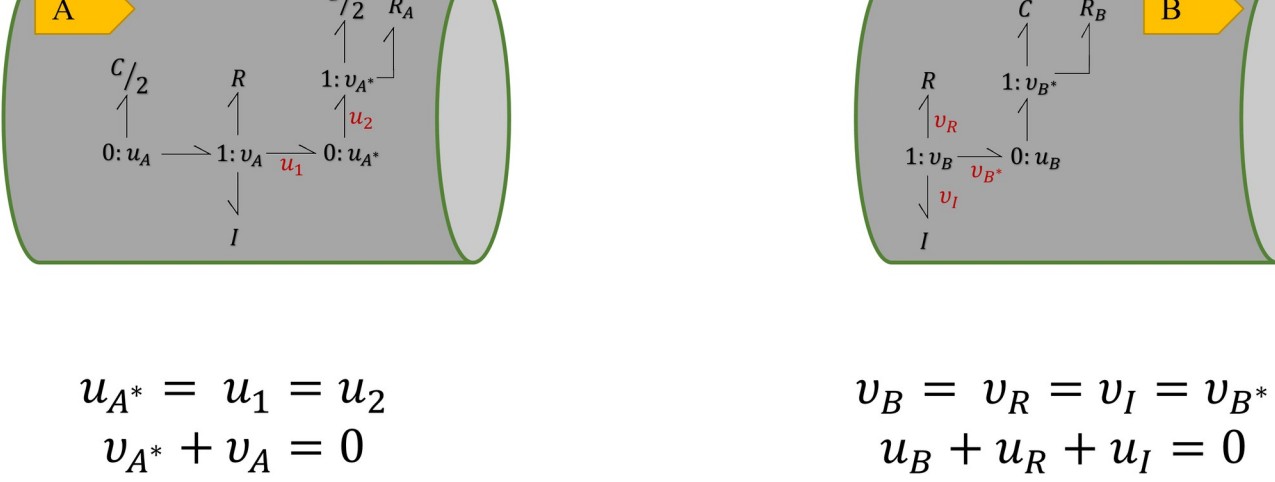

$$u_{A^*} = u_1 = u_2$$
$$v_{A^*} + v_A = 0$$

$$v_B = v_R = v_I = v_{B^*}$$
$$u_B + u_R + u_I = 0$$

**Fig 4. Conservation equations at the terminal junctions in two bond graph modules.** Equal flows at $1:v$ junction and equal potentials at $0:u$ junction. The sum of potentials equals to the inward potential in the $1:v$ junctions and the sum of flows equals to the outward flow in the $0:u$ junctions.

To track the changes to the constitutive equations, we have selected the $C/2$ and $R_A$ components at $1 : v_{A^*}$ junction in **A**. The constitutive equations in the case of Fig 4 would be:

$$\begin{cases} u_{R_A} = R_A v_{A^*}, \\[2mm] u_{C_{/2}} = \dfrac{2}{C}\dfrac{dv_{A^*}}{dt}. \end{cases} \qquad (7)$$

where based on the conservation laws, $v_{A^*} = -v_A$.

By creating an additional bond between the $0 : u$ junction in **A** and the $1 : v$ junction in **B**, these two segments can be coupled together. As potential is fixed in a $0 : u$ junction, an additional bond to the junction means the addition of one flow term in Eq (5). In the same manner adding a bond to a $1 : v$ junction means an extra potential term in Eq (6). This is illustrated in Fig 5 where the amount of $v_B$ in **B** is added to the conservation equation at $0 : u_{A^*}$ junction in **A** and also the amount of $u_{A^*}$ in **A** is added to the conservation equation at $1 : v_B$ junction in **B**.

By adding a bond between the junctions, the constitutive equations of the components at $1 : v_{A^*}$ junction in **A** remain as in Eq (7). However, based on the conservation laws in Fig 5, $v_{A^*}$ would be calculated as $v_{A^*} = -v_A - v_B$.

In order to compose models from these modules, we need to impose these changes into the CellML models. To do this, we propose adding auxiliary variables ($v_x$ or $u_x$, depending on the type of the junction) to each module. These auxiliary variables are set to zero by default, so they do not have any impact while the model is running separately. However, when the modules are integrated, we can readily create a mapping between the terminal junctions' known variables ($v$ or $u$, depending on the type of the junction). This is demonstrated in Fig 6.

We also need to deal with the case when a vessel is split into two or more branches. We have already stated that for any number of branching or even non-branching vessels, we use a common type of bond graph configuration. In this case, the idea is to impose *a priori* a maximum number of branching occurring in a vessel of the network (say four). Then we create our template module having four $v_x$ variables (see Fig 7). This creates the generic modules we require for composing the full circulation model.

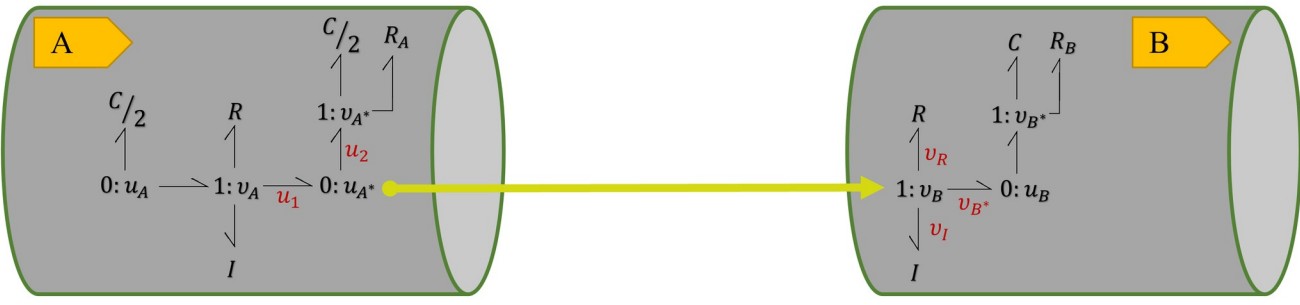

$$u_{A^*} = u_1 = u_2$$
$$v_{A^*} + v_A + \mathbf{v_B} = 0$$

$$v_B = v_R = v_I = v_{B^*}$$
$$u_B + u_R + u_I + \mathbf{u_{A^*}} = 0$$

**Fig 5. Conservation equations at the terminal junctions in two coupled bond graph modules.** $v_B$ in **B** is added to the conservation equation at $0 : u_{A^*}$ junction in **A** and $u_{A^*}$ in **A** is added to the conservation equation at $1 : v_B$ junction in **B**. Each terminal junction's $v$ or $u$ in each module is added to the other module's conservation equation, representing the addition of a bond.

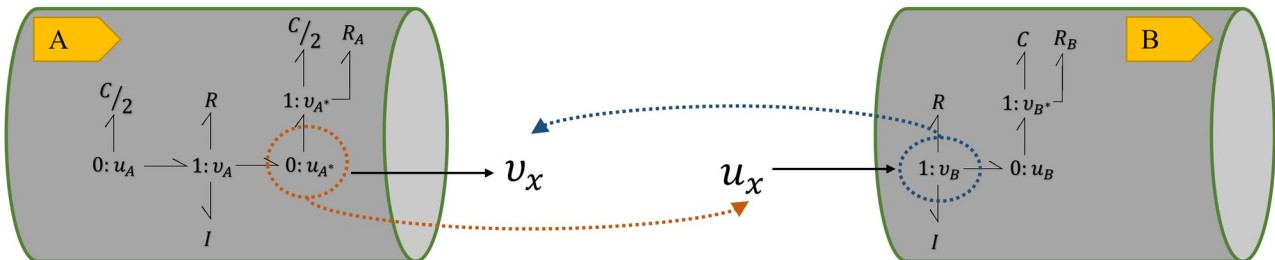

**Fig 6. Bond graph modular connection of the vessel models.** $v$ in the initial junction of the following module will be mapped to $v_x$ at the terminal junction of its preceding module. In the same way, $u$ at the terminal junction of each module will be mapped to $u_x$ at the initial junction of its following module. $v_x$ and $u_x$ are auxiliary variables.

## The SemGen merger tool

SemGen allows models encoded in CellML and SBML to be composed and is a powerful tool for semantics-based annotations, which enriches the models with the physiological and biological information about the entities and processes. Once modules are annotated, merging becomes easier as SemGen automatically detects the semantic overlaps and suggests to the user a list of identical or similar annotations in both modules. SemGen can merge only two models at one time. So, for the composition of *n* models, *(n-1)* merging steps are required. In this paper, we propose a hierarchical approach for module composition in fractal and repetitive segments in large systems, which reduces the number of merging tasks significantly. In the ADAN open-loop vascular model, the vessels can be grouped in seven subdivisions; HEAD AND NECK, HEART (cardiac output), TRUNK, LEFT ARM, RIGHT ARM, LEFT LEG and RIGHT LEG. Each subdivision can be created separately by merging the required vessel segments, and then be put together with other vascular sections. As the vascular segments for the limbs can be assumed identical, we can wrap the vascular model of a limb as a lumped module and then reuse it. The schematic of the created lumped modules of the ADAN open-loop model is illustrated in Fig 8. This lumping method can be generalised and applied to all types of systems that have similar sections.

The series of vessel segments in each lumped module is shown in Fig 9.

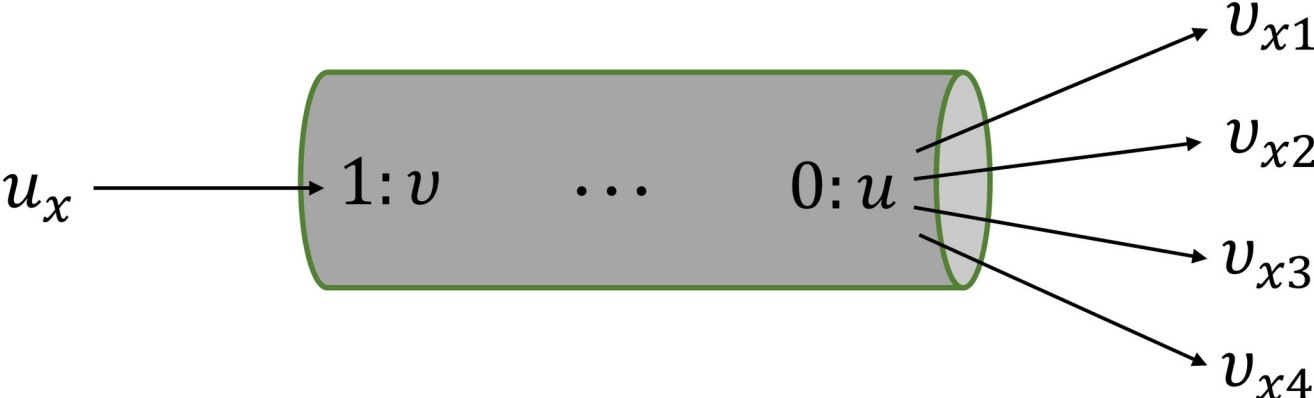

**Fig 7. The template bond graph module for intermediate vessels in our version of the ADAN open-loop model.** The number of $v_x$ variables depends on the maximum number of branches occurring in any segment. Here, four auxiliary variables corresponding to four branches are demonstrated.

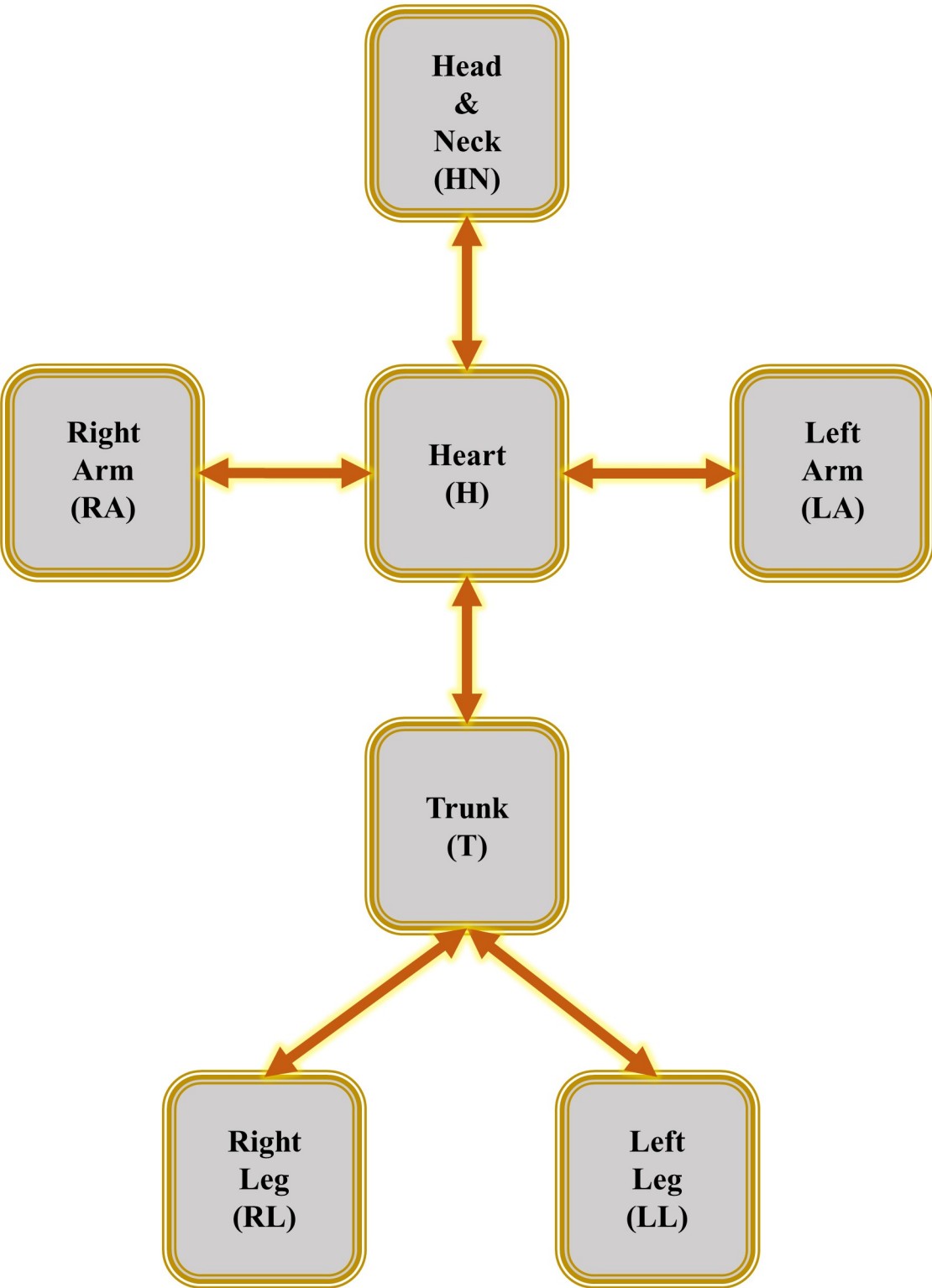

**Fig 8. Lumped modules in the ADAN open-loop model.** These lumped modules will be connected in the final step using the SemGen merger tool to produce the whole ADAN open-loop model.

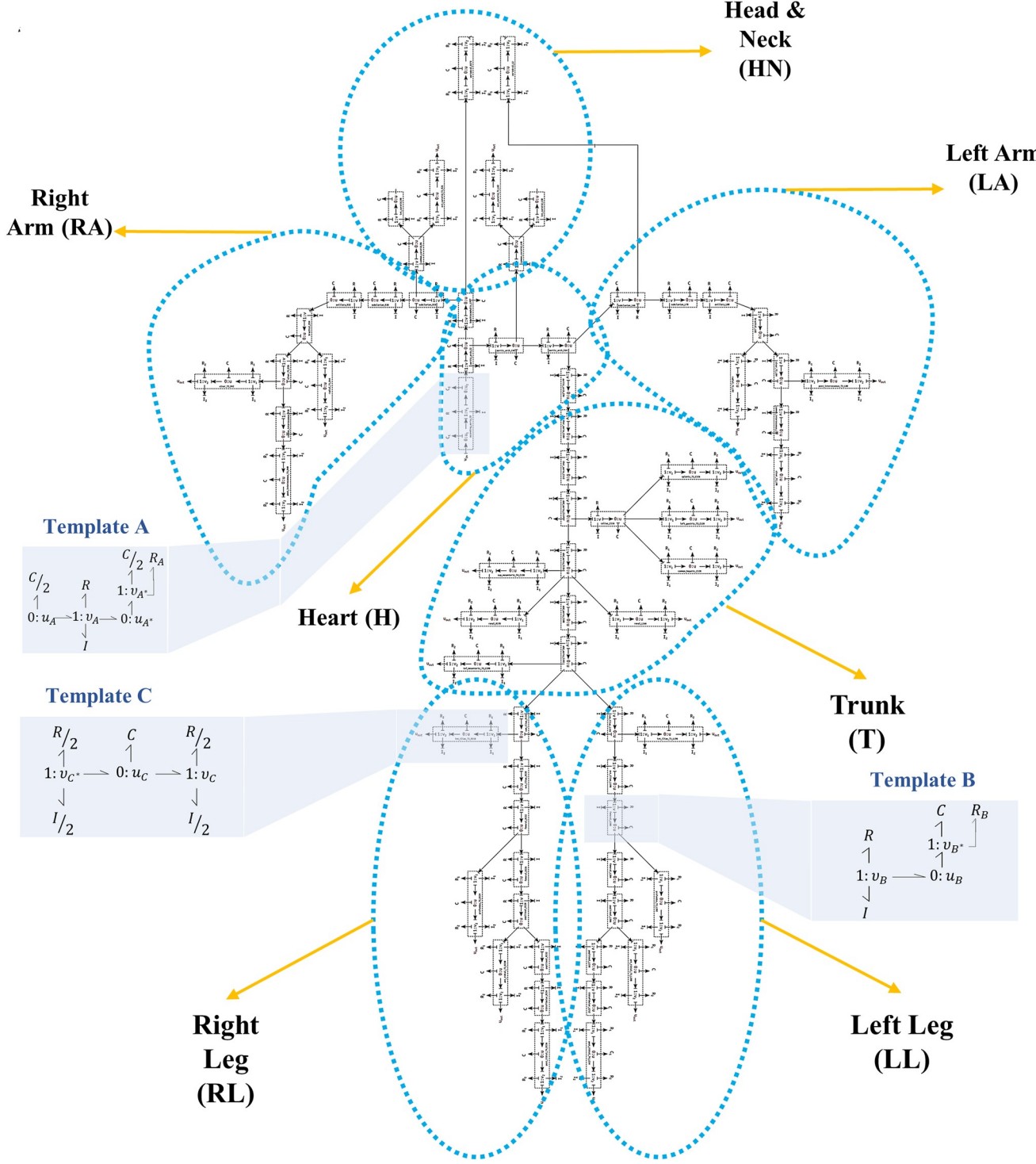

**Fig 9. Vessels network in the ADAN open-loop model.** The blue dashed shapes delineate the segments in each lumped module. The three bond graph templates of Fig 3 are shown in blue boxes for three exemplar segments. The segments network is adopted from [25].

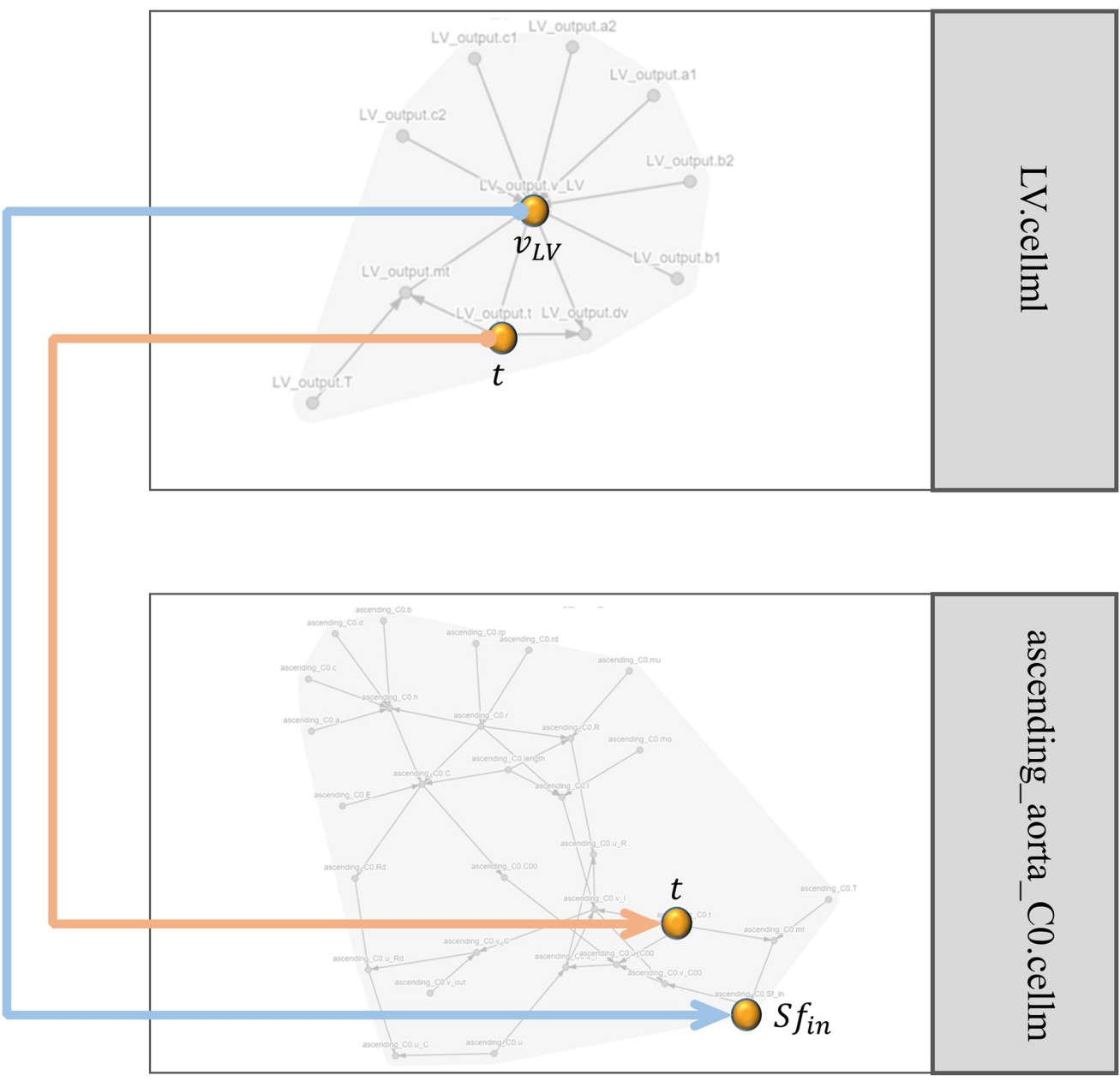

**Fig 10. The SemGen merger tool visualisation for two modules.** The model of left ventricle output flow (LV.cellml) is merged with the model of the ascending aorta (ascending_aorta_C0.cellml). Time (t) and the amount of the flow ($v_{LV}$) in the LV.cellml module are mapped to the time (t) and the source of flow ($Sf_{in}$) variables in the ascending_aorta_C0.cellml module. The grey lines show the internal dependencies between the variables in each module and the orange and blue lines show the links between the modules.

To visualise how the SemGen merger tool is utilised, the merging of two modules is depicted in Fig 10.

We started the model composition by creating the input to the system, which was generating the output flow of the heart left ventricle. The flow wave was obtained from digitising the analogous signal in the whole heart model, previously created in the ADAN closed-loop system [25]. The fitting task was performed in MATLAB, using a two-term Gaussian function as

in Eq (8) with the settings shown in S1 Table.

$$f(x) = \sum_{i=1}^{2} a_i e^{\left[-\left(\frac{x-b_i}{c_i}\right)^2\right]}$$

(8)

The digitised flow was imposed to both the newly merged model and the manually composed (ADAN open-loop) model in CellML for 10 seconds (S1 Fig).

   To compare the simulation results between the two approaches, normalised root mean square error (NRMSE) was also computed for each set of results as in Eq (9), where $\hat{y}i$ corresponds to the simulation points of our composed model using SemGen, and $y_i$ corresponds to the original ADAN open-loop simulation points. The normalisation was done relative to the difference of maximum and minimum data of the reference model (ADAN open-loop) in each simulation.

$$NRMSE = \frac{\sqrt{\sum_{i=1}^{n} \frac{(\hat{y}_i - y_i)^2}{n}}}{y_{max} - y_{min}}$$

(9)

## Results

We applied our hierarchical semantic model composition method to the bond graph ADAN model downloaded from the Cardiovascular Circulation Workspace on PMR (Physiome Model Repository). Thereafter, modules were extracted and modified to create templates (as described in Template modules section). Creating all the vessel segments from these templates, the modules where annotated and coupled using SemGen. The output model was then imported to OpenCOR (an environment to simulate CellML models: www.opencor.ws). The final composed model is available at https://github.com/Niloofar-Sh/ADAN-86-Bond-Graph-Model-Composition.

   The aim of our approach is to show that the ADAN model equations can be generated using automated semantic composition. Here, we compare the simulation results of our approach with the existing ADAN open-loop model. After reaching steady-state, blood flow and pressure were plotted from $t = 9$ s to $t = 10$ s. The results from the two models are compared in Figs 11 and 12. The plots show an almost exact match between the results of the two modelling approaches, indicating that the semantically composed model closely approximates the ADAN model. The minor differences that arise are due to the limbs' symmetrical nature, whereas the original ADAN model used separate parameters for the left and right limbs.

## Discussion

In this paper, we introduced a general method for assembling models from templated modules, demonstrated with the ADAN open-loop arterial model. This was done by encoding bond graph sub-models in CellML and using the annotation-based merging tool of SemGen to construct the system model using semantic annotations. Having the sub-models represented as bond graphs has allowed us to systematically compose a model from a large number of modules. Furthermore, enriching the modules with biological semantics assists us in coupling them. In the case where the system has similar sections consisting of several modules (here we have symmetric limbs in the ADAN open-loop model), a hierarchical approach to model composition can be utilised. Once a section is composed, other similar sections can be created by generating copies of that section.

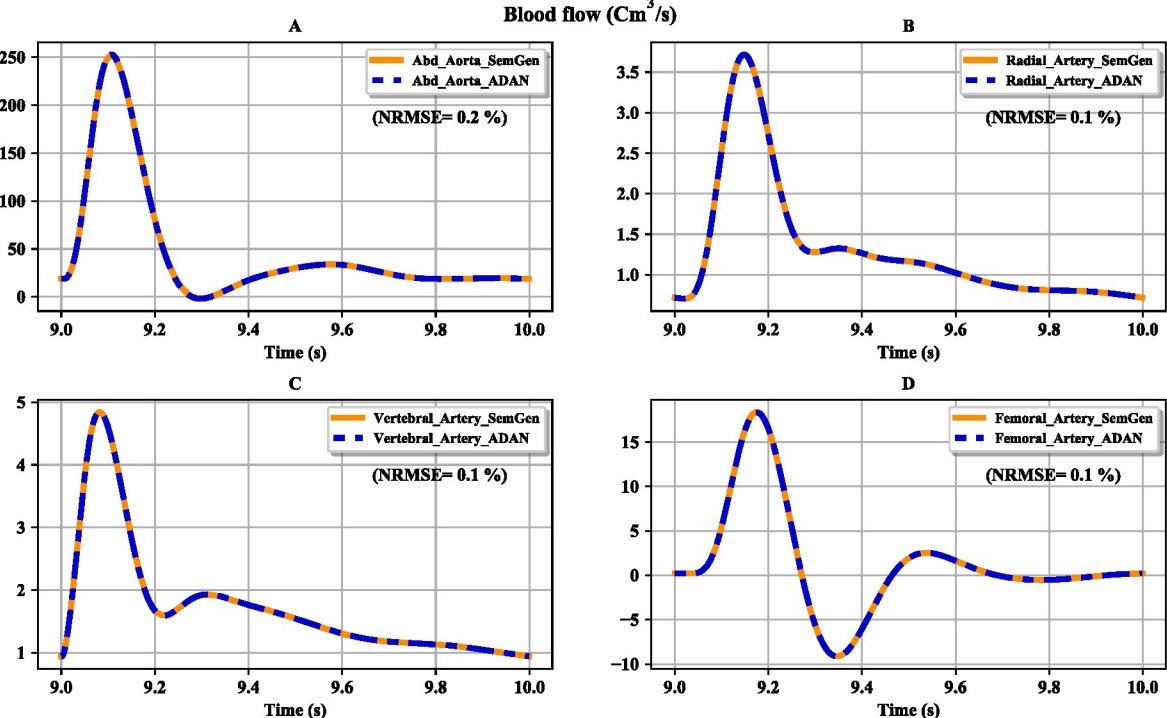

**Fig 11. Comparison between the flows obtained from the ADAN open-loop model and our version of the model.** (A) Flow in the abdominal aorta; (B) flow in the radial artery; (C) flow in the vertebral artery; (D) flow in the femoral artery. The normalised root mean square errors (NRMSE) are represented as percentages.

Simulation results were compared to the ADAN open-loop model. The slight differences in the flow and pressure of the vessels in the two models originate from the fact that unlike the original ADAN model, similar limbs were considered identical. Supposing that similar limbs are identical and exactly have the same vessel segments and parameter values, the modules for one limb were merged, and this integrated model was copied and used for the other limb as well (both for arms and legs). This significantly accelerated the process of merging models at the expense of minor differences in simulation results. Even if this simplification is not made and each limb segments are coupled individually, still the model composition procedure remains fast and more reliable than editing the CellML code directly in complex biological systems.

Currently, SemGen does not allow importing other CellML models to the main model; for example, in [25], the list of required units (*Units.cellml* module) was imported to the *main.cellml* model. Thus, in our recreated model, all the required units were defined in the same file in which each module was defined. We expect this file importing feature will be added to Sem-Gen in the near future. Although the current composition method is more convenient and faster compared to time-consuming and error-prone manual merging, it does require adding extra variables to the modules and managing the mappings between the biological annotations in SemGen. According to the long-term goals presented for SemGen, the capability of equation modification in the case of merging two models will be added [49]. This will enable the reuse of the existing modules without needing to add such extra variables.

This approach can also be utilised in other domains of application, for example cellular biochemistry, where different modules of biochemical reactions couple together and represent a biological pathway or network. Bearing in mind that one biochemical species can be both

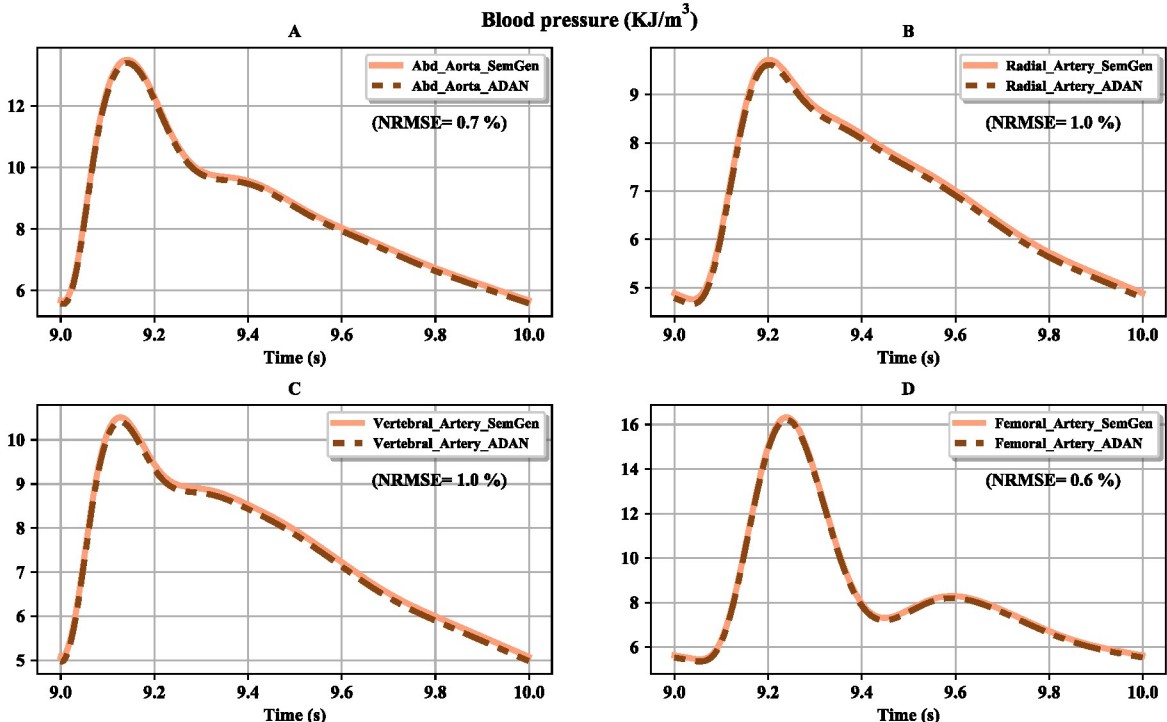

**Fig 12. Comparison between the pressures obtained from the ADAN open-loop model and our version of the model.** (A) Pressure in the abdominal aorta; (B) pressure in the radial artery; (C) pressure in the vertebral artery; (D) pressure in the femoral artery. The normalised root mean square errors (NRMSE) are represented as percentages.

produced in one reaction and consumed by another reaction, the rate of molar change for each species would be the summation of multiple reaction rates. By adding and annotating auxiliary variables to the constitutive equations in each module, a number of ports for model composition will be available. The idea of manual 'white box' composition of bond graph models of biochemical systems has previously been explored by Gawthrop [20] where a modular bond graph model of mitochondrial electron transport chain was developed. Gawthrop et al. [58] have also described a reusable and modular model of glycogenolysis in skeletal muscle. Our work expands on these approaches through the addition of the biological semantics, which enables the automation of model composition.

The ADAN closed-loop system (including the cerebral arteries network with 42 further vessel segments) can also be generated using this approach. Arterio-arterial anastomoses, where the arteries merge into one, are not included in the ADAN open-loop model, thus the trunk and limb arteries in our model merely bifurcate and expand as they move distally. Unlike the limbs in our model, the cerebral arteries merge at several points (for instance where the right and left hemispheres are connected). To incorporate this structure, we need an additional template module (four modules in total). Comparable to the limbs which we assumed have identical vascular segments, our lumping method can be applied to brain hemispheres in the same manner.

Bond graphs are restricted to lumped-parameter models and are not naturally applicable to continuous PDEs. However, bond graphs can be represented in a more generalised formulation, known as Port-Hamiltonians [59], in which models can be described in PDEs.

## Conclusion

We have utilised the the SemGen annotator and merger tools to create a composed model of artery segments based on the ADAN open-loop model. Describing the vessel segments in terms of bond graphs helped us avoid post-merging adjustments in the model equations (as is almost always necessary in dealing with conventional computational models). The bond graph framework facilitates the procedure of extending biological and physiological models with minimal error and effort. Also, using the SemGen merger tool for coupling the modules allowed us to skip the manual code-wise mappings between the modules in CellML. We anticipate that our approach will enable future work on constructing multiscale models of organs that bridge biochemical processes at the cellular level to tissue-level processes such as circulation.

## Supporting information

**S1 Table. MATLAB fitting tool settings and estimated parameters in digitising the cardiac output flow.** The fitting options which are not mentioned in the table are remained untouched as their default values in the MATLAB fitting tool.
(PDF)

**S1 Fig. The digitised cardiac output flow.** The yellow line and dashed red line show the same imposed flow in the SemGen merged model and the ADAN open-loop model.
(PDF)

## Acknowledgments

NS would like to thank Maxwell Neal, Keri Moyle, and Anand Rampadarath for their helpful comments and suggestions.

## Author Contributions

**Conceptualization:** Niloofar Shahidi, Soroush Safaei, Edmund J. Crampin, David P. Nickerson.

**Data curation:** Niloofar Shahidi.

**Formal analysis:** Niloofar Shahidi.

**Funding acquisition:** Edmund J. Crampin, David P. Nickerson.

**Investigation:** Niloofar Shahidi, Soroush Safaei.

**Methodology:** Niloofar Shahidi.

**Project administration:** Kenneth Tran, Edmund J. Crampin, David P. Nickerson.

**Resources:** Edmund J. Crampin, David P. Nickerson.

**Software:** Niloofar Shahidi.

**Supervision:** David P. Nickerson.

**Validation:** Niloofar Shahidi, Michael Pan, Soroush Safaei, Kenneth Tran, Edmund J. Crampin, David P. Nickerson.

**Visualization:** Niloofar Shahidi, Michael Pan, Kenneth Tran, Edmund J. Crampin, David P. Nickerson.

**Writing – original draft:** Niloofar Shahidi.

**Writing – review & editing:** Niloofar Shahidi, Michael Pan, Soroush Safaei, Kenneth Tran, Edmund J. Crampin, David P. Nickerson.

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
