## [Decision Letter · Decision Letter 0]

14 Apr 2021

Dear Shahidi,

Thank you very much for submitting your manuscript "Hierarchical semantic composition of biosimulation models using bond graphs" for consideration at PLOS Computational Biology. As with all papers reviewed by the journal, your manuscript was reviewed by members of the editorial board and by several independent reviewers. The reviewers appreciated the attention to an important topic. Based on the reviews, we are likely to accept this manuscript for publication, providing that you modify the manuscript according to the review recommendations.

I point out that one of the reviewers, in comments to the editor, expressed some skepticism about the utility of the bond graph approach in the broader biomodeling community. Your revised paper might consider more discussion towards this skepticism. As a suggestion, you might briefly review other software modeling tools that adopt the bondgraph formalism. Examples include the Modelica and the commercial Mathworks Physical Modeling Toolbox.

Sincerely,

Daniel A Beard

Deputy Editor

PLOS Computational Biology

[LINK]

Reviewer's Responses to Questions

**Comments to the Authors:**

Reviewer #1: No comments

Reviewer #2: The paper addresses a general methodology for composing models, combining the energy-based bond graph approach with semantics-based annotations. This approach is proposed for biological / physiological models and ensures that the composite model is physically plausible. The technique is applied to automated model composition using a model of human arterial circulation.

The contributions of the paper are very interesting and promising. The paper is written in a comprehensive manner and it has technical soundness. The references are appropriate. All in all, this is a very good paper.

Some minor issues:

1. It would be interesting for the readers to discuss some aspects regarding the use of entire arterial network (i.e., including the cerebral system). Which are the difficulties and problems that could arise when we deal with the cerebral system?

2. The overall model is based on lumped-parameter models. There are some losses when we use this kind of model instead of PDEs description? Is there a workable solution with bond graphs for biological systems modelled via PDEs?

**Have the authors made all data and (if applicable) computational code underlying the findings in their manuscript fully available?**

Reviewer #1: Yes

Reviewer #2: None

PLOS authors have the option to publish the peer review history of their article (what does this mean?). If published, this will include your full peer review and any attached files.

Reviewer #1: No

Reviewer #2: No

Figure Files:

Data Requirements:

Reproducibility:

References:

---

## [Editor Report · Decision Letter 1]

27 Apr 2021

Dear Shahidi,

We are pleased to inform you that your manuscript 'Hierarchical semantic composition of biosimulation models using bond graphs' has been provisionally accepted for publication in PLOS Computational Biology.

Best regards,

Daniel A Beard

Deputy Editor

PLOS Computational Biology

Daniel Beard

Deputy Editor

PLOS Computational Biology

---

## [Editor Report · Acceptance letter]

10 May 2021

PCOMPBIOL-D-21-00472R1 

Hierarchical semantic composition of biosimulation models using bond graphs

Dear Dr Shahidi,

I am pleased to inform you that your manuscript has been formally accepted for publication in PLOS Computational Biology. Your manuscript is now with our production department and you will be notified of the publication date in due course.

With kind regards,

Andrea Szabo
